# Multibody Models of the Thoracolumbar Spine: A Review on Applications, Limitations, and Challenges

**DOI:** 10.3390/bioengineering10020202

**Published:** 2023-02-03

**Authors:** Tanja Lerchl, Kati Nispel, Thomas Baum, Jannis Bodden, Veit Senner, Jan S. Kirschke

**Affiliations:** 1Sport Equipment and Sport Materials, School of Engineering and Design, Technical University of Munich, 85748 Garching, Germany; 2Department of Diagnostic and Interventional Neuroradiology, School of Medicine, Klinikum Rechts der Isar, Technical University of Munich, 81675 Munich, Germany

**Keywords:** musculoskeletal multibody dynamics, spinal biomechanics, spinal alignment, spinal loading, muscle force computation, thoracolumbar spine, biomechanical model

## Abstract

Numerical models of the musculoskeletal system as investigative tools are an integral part of biomechanical and clinical research. While finite element modeling is primarily suitable for the examination of deformation states and internal stresses in flexible bodies, multibody modeling is based on the assumption of rigid bodies, that are connected via joints and flexible elements. This simplification allows the consideration of biomechanical systems from a holistic perspective and thus takes into account multiple influencing factors of mechanical loads. Being the source of major health issues worldwide, the human spine is subject to a variety of studies using these models to investigate and understand healthy and pathological biomechanics of the upper body. In this review, we summarize the current state-of-the-art literature on multibody models of the thoracolumbar spine and identify limitations and challenges related to current modeling approaches.

## 1. Introduction

Chronic back pain is one of the major health issues worldwide. Though general risk factors such as occupation, obesity or anthropometric parameters could be identified in the past years [1], the specification of individual biomechanical indicators for the prediction of symptoms and chronicity is challenging, as it requires an in-depth knowledge of spinal kinematics and resulting loads. Even though experimental methods are essential to help build this knowledge, they come with limitations. In vitro studies can help understand segment mechanics but are not applicable when it comes to the investigation of complex in vivo biomechanics of the whole torso [2]. The invasive character of the in vivo measurement of these parameters via intradiscal pressure sensors [3,4] or instrumented vertebral implants [5,6] makes these methods unsuitable for clinical analysis. Computational, biomechanical models can provide a valuable alternative when it comes to the estimation of spinal loads. There are two approaches for the numerical analysis of spinal loading. While finite element models (FEM) hold the potential to investigate internal stress states in flexible bodies and their underlying or resulting deformation, multibody models (multibody system, MBS) can help analyze mechanical loads on the musculoskeletal system at a holistic level. Breaking the system down to its essential mechanical components, classic MBS models incorporate rigid bodies connected by joints and, depending on the respective research question, force elements representing flexible structures such as intervertebral discs (IVD), ligaments, cartilage, and other connective tissue. This way, MBS models represent a valuable tool to increase a profound understanding of healthy and pathological biomechanics. Gould et al. published a review on FEM and MBS models of the healthy and scoliotic spine in 2021 [7]. Focusing on the latter one, the authors state that their review provides solely a brief overview on MBS models of the healthy spine and refer the reader to the review on MBS modeling of the cervical spine by Alizadeh et al. [8] and the review by Dreischarf et al. on in vivo studies and computational models, published in 2016 [9].

The wide range of applications, improved technical capabilities, and increasing knowledge of spinal biomechanics, which answer old questions and raise new ones, mean that the demand for high-quality MBS models is not abating. As a consequence, the number of published models is increasing every year providing new opportunities and insight.

In recent years, models have been introduced that extend the classic notion of a multibody or musculoskeletal models. These models incorporate flexible bodies such as beam elements into rigid body models and thus soften the boundary between FEM and MBS models [10,11]. However, within the scope of this work, we want to review the developments in the field of multibody models of the healthy thoracolumbar spine, focusing on classical rigid body models. Hereby, we shed light on common modeling methods and applications, as well as identify and discuss related limitations and challenges in state-of-the-art spine modeling.

## 2. Methods

To generate a list of potentially relevant publications, a systematic search was carried out in PubMed and Scopus in November and December 2022. The search included the keywords “spine AND model AND ((multi AND body) OR musculoskeletal)”. Excluding results prior to 2013 left 1288 publications on PubMed and 1304 on Scopus. However, relevant citations in the articles were also included, if they were published before 2013. Subsequently, duplicates were removed by identical PubMedIDs and titles. Remaining articles were then filtered by title and abstract and the full text eventually analyzed. Publications were excluded if they featured at least one of the following topics:Finite element modeling;Models of the cervical spine;Models without muscle incorporation;Models of the scoliotic spine;Models of the nonhuman spine;Studies with a medical scope other than biomechanics.

Inclusion criteria were set to
Musculoskeletal models;Multibody models;Models of the thoracolumbar spine;Models of the healthy spine.

We analyzed the remaining studies systematically according to the represented modeling methods and applications and identified existing limitations and challenges.

## 3. Multibody Modeling of the Healthy Spine

After filtering a total of 2592 articles, 81 articles remained, which were included in this review. Focusing on extensive musculoskeletal models of the thoracolumbar spine, we discuss models with reduced complexity, such as abstracted models [12,13,14,15,16], skeletal models neglecting muscular effects [17,18] or models of the lumbar spine [19,20,21,22,23,24,25,26,27,28,29] only in passing.

Overall, our literature review revealed that a large proportion of published studies was based on a few original models [30,31,32,33]. Due to the accessibility of these models via the commercially available software AnyBody (AnyBody Technology A/S, Aalborg, Denmark) [30,33] or the open-source software OpenSim [31,32,34], numerous studies can be found that used, modified, and extended these models, beyond the boundaries of the respective research groups as well [35,36,37,38,39,40,41,42,43,44,45,46,47,48,49,50,51,52,53,54,55,56,57,58]. Apart from these widely reused models, further original models can be found in the literature using alternative software [59,60,61,62,63,64]. Table 1 provides an overview of the original models found and subsequent studies associated with them.

### 3.1. General Model Setup and Kinematics

In the past two decades, simplified models of the whole torso with a detailed lumbar spine were developed to investigate lumbar loads [30,31,59,61,69]. One of the first generic models for lumbar load estimation was introduced by de Zee et al. in 2007 [30], which comprised seven rigid bodies for the pelvis including the sacrum, five lumbar vertebrae, and one lumped segment representing the thoracic spine including the rib cage and cervical spine. The model anatomy was based on publications by Hansen et al. [78] and Bodguk et al. [79]. De Zee defined intervertebral joints as spherical joints with their respective center of rotation (COR) located in the intersection of the instantaneous axis of rotation and the midsagittal plane according to Pearcy and Bodguk [80]. A total of 154 actuators representing muscle fascicles for the erector spinae (ES), rectus abdominis (RA), internal obliques (IO), external obliques (EO), psoas major (PM), quadratus lumborum (QL), and multifidus (MF) were implemented in the model either as a straight line between insertion and origin or, in order to mimic more realistic lines of action, redirected using so-called via points or wrapping surfaces [30].

Inspired by de Zee’s model, Christophy et al. published a generic multibody model of the lumbar spine in 2012 [31], incorporating a more detailed muscle architecture regarding the latissimus dorsi (LD) and the MF muscle. Using the open-source software OpenSim [34], the model has been widely used and extended in the past years [31,37,40,41,48,49,50,51,52,81,82]. In recent years, other models with simplified thorax have been published [59,61,69].

Favier et al. published a full-body model with a detailed lumbar spine in 2021 [69]. The model was created in OpenSim and included in total 20 rigid bodies including the head–neck, three-segment thoracic and cervical spine (spherical joints in T7-T8 and C7-T1), five lumbar vertebrae, pelvis with sacrum, as well as upper and lower extremities. The model incorporated a total of 538 muscle actuators for the lower limbs and lumbar spine [69].

Lerchl et al. introduced a pipeline for the semiautomated generation of individualized MBS models with a detailed lumbar spine created in the commercial multibody modeling software Simpack (Dassault Systèmes, France) in 2022 [59]. Based on CT data, the models included individual vertebrae T1-L5 with a fused thoracic part and rib cage and spherical lumbar intervertebral joints, and generic segments for the head–neck, pelvis, sacrum, and simplified arms. A total number of 103 actuators representing the muscles of the lower back were incorporated [59].

Research devoted to the loading of the thoracic spine is less common and therefore, only few models incorporating a detailed thoracic spine and rib cage can be found in the literature [32,33,72]. As opposed to musculoskeletal models with a rigid thorax, these models allow a comprehensive analysis of spinal loading for load cases involving thoracic movement. Based on the generic model of the lumbar spine by de Zee et al. [30], Ignasiak et al. introduced a musculoskeletal model of the thoracolumbar spine with a detailed articulated rib cage [33]. Ignasiak et al. extended the model by individual rigid bodies of 12 vertebrae, 10 pairs of ribs, and a sternum. Intervertebral thoracic joints were defined as six-DOF joints and lumbar joints, originally modeled as spherical joints [30], were also modified, respectively. Costovertebral (CV) and costotransverse (CT) joints were defined as revolute joints with the rotation axis in the frontal direction and all joints between the ribs and the sternum were modeled with six DOFs, except the first pair, which were modeled as spherical joints. The model was validated against in vivo data and used in follow-up studies [33,39,67,68].

A comprehensive model of the upper body including 60 segments (vertebrae, ribs, skull, sternum, hyoid, thyrohyoid, clavicles, scapulas, humeri, sacrum, and pelvis) created in AnyBody was published by Bayoglu et al. in 2019 [72].

Based on the lumbar spine model of Christophy et al. [31], Bruno et al. developed and validated a fully articulated model of the thoracolumbar spine in OPENSIM including individual vertebrae, ribs, and sternum [32]. Like Christophy’s model, the thoracolumbar model of Bruno et al. has been widely used and adapted since its publication [32,43,54,56,57,58,83,84].

In biomechanical MBS modeling, intersegmental connections are usually implemented as joints with defined DOFs, which can either be defined directly in the joint or are implemented as constraints, limiting the joint’s effective degrees of freedom to its relevant components. It is common practice to model intervertebral joints as spherical joints allowing rotation around three spatial axes [31,62]. Few models exist, that defined intervertebral joints with six DOFs, additionally accounting for translational motion [33,37,41,61]. The centers of rotation are located either in the geometrical center of the IVD [33,59,62] or in the instantaneous axis of rotation according to Pearcy and Bodguk [30,31,69,80]. CV joints are modeled as pin joints rotating around the vector between the costovertebral and costotransverse joints [32,33,72] or spherical joints [64] and CS joints as six DOFs [33,64,72]. Depending on the simulation approach (Section 3.4), kinematic data have been most commonly assigned to the respective DOFs according to findings from our own experimental studies or the literature (Section 3.3). This way, model kinematics are usually described using relative minimum coordinates. However, for inverse kinematic approaches, absolute coordinates are assigned to the end link of the kinematic chain. Providing stable boundary conditions for the mechanical analyses, the models are usually connected to the inertial frame of reference and therefore leaving the head–neck complex as the end link of the open kinematic chain. Upper-body weight is either combined and included in the center of mass of the lumped thoracic body [61], distributed according to the literature [85,86] or derived from patient-specific CT or MRI data and distributed levelwise along the thoracolumbar spine [59,62].

### 3.2. Passive (Visco)elastic Components

Various approaches have been taken regarding the modeling of viscoelastic structures that passively stabilize the spine, such as IVD, spinal ligaments, or the (cartilage) tissue of the thorax. The modeling approach can vary both in the level of detail and in the mechanical characteristics considered. Thus, some models neglect the effects of these components entirely [30,31,32,62], whereas others combine them partially or completely into one single stabilizing element per joint [60,69,72], or even integrate individual components explicitly [59,61,64,77]. The majority of approaches simplify the mechanical properties of connective tissue to linear elastic force elements, which produce corresponding forces and moments exclusively depending on their deformation. In multibody models, such material behavior is described via spring elements with constant stiffness for the corresponding DOFs. Only a few models incorporate the nonlinear mechanical behavior of biological passive structures [87]. However, modeling these components as purely elastic does not account for viscous effects that influence the mechanical response as a function of the deformation rate, also known as damping behavior. A detailed nonlinear viscoelastic modeling of IVDs and spinal ligaments, such as the anterior and posterior longitudinal ligament, the flavum ligament, as well as the interspinal and supraspinal ligament, can be found in only a few models [59,61]. The respective parameters are usually taken from in vitro studies available in the literature [88,89,90,91,92].

To examine thoracic loads, models require an appropriate force transmission from the rib cage to the thoracic spine in addition to intervertebral passive structures. In this context, costosternal (CS), costotransverse (CT) and costovertebral (CV) articulations are a central issue. Commonly, these connections are constrained and modeled as linear elastic elements according to the resulting DOFs. Stiffness parameters are usually taken from in vitro studies or adapted from previously published in silico studies. Bruno et al. included point-to-point actuators, which were placed between the ends of the ribs and the sternum (ribs 1–7) or between the ends of adjacent ribs (ribs 8–10) to represent forces transmitted by costal cartilage. As a result of a sensitivity analysis, forces generated by the actuators were set to 1000 N allowing the costal cartilage to provide a high supporting force to the end of the ribs [32].

Mechanical properties are usually incorporated either directly from mechanical testing, such as ligament tensile tests [88,93] or by simulating in vitro protocols, such as stepwise reduction studies, where individual connective structures are gradually removed from functional biological units, such as the FSU or the rib cage, while measuring the mechanical properties of the units after every resection [89,94,95]. However, due to the high level of intra- and interindividual variability regarding the mechanical characteristics of biological materials, the resulting parameters usually come with high standard deviations [96].

### 3.3. Scaling and Individualization

Spinal loading is highly dependent on a variety of subject-specific characteristics, such as spinal alignment, anthropometry, body weight distribution, or kinematics. While finite element models exist that account for individual characteristics [97,98,99,100,101,102,103,104], multibody models are predominantly generic in nature. In the past years, an increasing number of studies have been published, putting an emphasis on the individualization of the models [54,55,58,59,62].

A wide range of MBS models are based on measurements available in respective databases, e.g., in the OpenSim database (https://simtk.org/projects/osimdatabase, accessed on 27 December 2022). To gain reliable insights for the examined load cases, it is important to match the subject characteristics to the investigated kinematics as congruently as possible. It is common scientific practice to use available data based on measurements of bone geometries derived from imaging data or cadaver studies of individuals and scale and adapt the relevant parameters to the desired anthropometry depending on the characteristics of the studied target group. The need to make use of various sources in this regard makes it essential to be clear about the underlying data sets, in order to draw meaningful conclusions from simulation results. Thus, segment masses and body weight distribution and simplified kinematics are usually taken from the literature [85,86,105]. Some studies include experimental data collection of kinematics to scale the existing model appropriately [45,51,83] and include muscle activity from electromyography (EMG) measurement to drive the model [52]. This usually does not incorporate individual bone geometries, muscle morphology, or the mechanical properties of viscoelastic components.

However, the neglect or only limited consideration of interindividual variation makes these models poorly suitable for a detailed subject-specific analysis. Models based on coherent datasets regarding bone geometry, anthropometry, and muscle architecture, and kinematics are rare in the literature. Bayoglu et al. built a model based on extensive measurements of one cadaver, incorporating general kinematic data from the literature [72,73,74].

Dao et al. published a patient-specific model based on CT and MRI data [20] of the lumbar spine. Bruno et al. used their generic model [32] for the investigation of the impact of the integration of subject-specific properties [42]. Therefore, they incorporated CT-based measurements of trunk anatomy, such as spinal alignment and muscle morphology, indicating the relevance of considering these factors [42]. Based on this publication, Banks et al. investigated lumbar load in a patient-specific MBS model using CT data and marker-based motion capturing to combine individual musculoskeletal geometry and coherent kinematics [58]. However, the individualization of those models usually involves a time-consuming, manual, or semiautomated process which requires expert knowledge. To the best of our knowledge, only two publications can be found that deal with the topic of automating the individualization of MBS models [59,62].

Fasser at al. used annotated bi-planar radiography images (EOS imaging, Paris, France) for the automated generation of semi-subject-specific MBS models of the torso. The models included individual size and the alignment of bony structures as well as an individual body mass distribution. In the process, 112 and 109 points were marked in the frontal and sagittal plane, respectively, and converted into 3D coordinates. The body mass distribution was determined using the individual body contour of the imaging data. Individual bone geometries, muscle morphology, and passive elements were not included in the model [62].

Based on the use of artificial neural networks (ANN), Lerchl et al. introduced a pipeline for the automated segmentation of vertebrae [106] and soft tissue of the torso, as well as the generation of the points of interest defining muscles and ligaments’ attachment points and the location and orientation of intervertebral joints. All data were derived from CT imaging and the model generation required minimal manual interaction, making it suitable for the analysis of large patient cohorts. However, the individual characteristics of the muscles and connective tissue could not yet be integrated in the process [59].

### 3.4. Muscle Force Estimation

A mechanical analysis with multibody systems can follow two approaches, which define the necessary input data. Forward dynamic simulations (FD) require kinetic data to drive the model to generate specific kinematics. This usually means that muscle forces are applied directly or indirectly to the model to produce a desired motion. This is contrasted with the idea of inverse dynamic simulations(ID), which use kinematic data as input to calculate the required kinetic data. Thus, joint kinematics during as specific movement is imposed to the model and necessary joint moments and therefore, associated muscle forces are calculated. However, having more control variables, namely, muscle fascicles, than DOFs, the human musculoskeletal system is redundant. This leads to an infinite number of solutions for each load case. In order to determine the most suitable solutions, a mathematical optimization is a commonly used method. Numerous algorithms are available to find the optimal solution. Hereby, depending on the chosen algorithm, control variables, namely, muscle activation, excitation, or forces are varied in a deterministic or stochastic way until some given optimality criteria and constraints are met. Most commonly, a combination of inverse dynamics and static optimization (SO) is used [30,32,45], sometimes including inverse kinematics (IK) to determine individual joint kinematics [62,63,69]. The inverse dynamic simulation provides joint moments necessary to generate the simulated movement. Subsequently, the static optimization solves the redundancy problem for each time frame sequentially under the consideration of meeting equilibrium conditions.

In MBS models of the spine, muscles of interest are usually modeled as multiple fascicles, which comprehensively consider the respective lines of action (Section 3.1). Individual fascicles are modeled either as simple force actuators or, more complex, as Hill type muscles [107]. The classic muscle model according to Hill comprises serial and parallel elastic elements, representing passive elastic properties of the muscle–tendon complex as well as a contractile element representing the active component, namely, the function of myofilaments. This element can include muscle-specific characteristics, such as the force–length and force–velocity relationship as well as activation dynamics. Depending on how far these dynamics are taken into account, the muscle excitation, activation, or force can drive the model and therefore represent control variables for optimization routines. Detailed definitions of muscle-specific dynamics can be found in the literature [108,109].

## 4. Applications of MBS Models

MBS models can be used to address a wide range of questions. There are numerous publications devoted to the evaluation of methods in numerical modeling, including sensitivity analyss or validation studies. Furthermore, validated models can help to gain valuable insights into biomechanically or clinically relevant load cases. However, depending on the investigated load case and subject collective, model extensions, and modifications are usually necessary. Table 2 provides an overview of the most relevant studies using existing models to address specific research questions.

### 4.1. Studies with Methodological Focus

Various publications can be found in the literature evaluating and validating new approaches in MBS modeling [19,30,31,32,45,63,64,69]. For the purpose of validating these approaches, it is common scientific practice to compare simulation results with existing results from in vivo or in vitro measurements. Of note, those comparisons are mainly relative, as few in vivo measurements are available and exact boundary conditions are hard to control. Frequently used in vivo studies to validate results on spinal loading from simulation are intradiscal pressure measurements [4,114]. Estimated muscle forces are usually compared to EMG measurements from one’s own experimental studies [48] or the literature [59].

Apart from evaluating the validity of the modeling approach, the simulation results of generated MBS models can be used to validate novel methods in data processing regarding the derivation of both relevant modeling data from imaging [19,20,21] and kinematic data motion capture [54]. Due to the usually extensive effort connected to the processing of individual data, recent publications have focused on the automation of the process [59,62].

Simplifications are an integral part of any model and have to be taken into consideration when it comes to the interpretation of the results. To understand and evaluate their influence, MBS models have been used to systematically investigate common assumptions, such as the reduction of complex mechanics of the functional spine unit (FSU) [37,115]. Further, the sensitivity of the model accuracy to assumed positions of intervertebral centers of rotation [23,36] or muscle insertions [75] have been analyzed. Rockenfeller et al. investigated the effect of muscle- or torque-driven centrodes using an MBS model of the lumbar spine.

Furthermore, a systematic model-based analysis can help standardize clinical procedures, such as the classification of spinal shapes [116] or to define boundary conditions for experimental protocols [24].

### 4.2. Studies with Biomechanical or Clinical Focus

Validated models are used to comprehensively investigate biomechanical and clinical aspects of a wide range from routine scenarios to nonphysiological, or even traumatic events.

The relevance of low-dynamic everyday or work-related activities for the general population, as well as their experimental accessibility, make these scenarios among the most studied in biomechanical simulations. Therefore, numerous models exist that deal with the mechanical effects of lifting [12,13,25,46,76,77,82], everyday activities such as walking, flexion, extension, or lateral bending [15,43,69] or work-related situations such as high-frequency axial loading [17,18]. In this context, different lifting techniques were evaluated [50,51,83,117]. Accident situations were investigated by Wei et al. [16] for snowboarding and for frontal impact by Valdano et al. [14]. Incorporating noncritical higher dynamics, Raabe et al. combined a generic model of the lumbar spine [31] with a model of the lower limbs [111] to analyze the biomechanics of jogging [40]. Studies investigating specific kinematic boundary conditions usually involve an experimental setup to collect kinematic data in a healthy adult population [46,47,52,58,83]. Comparably few studies target more vulnerable populations, such as amputees [48,53] or children [27,56], who used validated models of adults and scales them according to the literature to match the average anthropometric data of children.

Regarding the influence of healthy anatomical and anthropometric and anatomical characteristics, biomechanical modeling have been used to determine the effect of spinal alignment [28,43,47], to gain insight into load sharing of passive structures of the FSU [22], the effect of ligament stiffness [65] or muscle strengthening [118].

Furthermore, MBS models can help to understand and treat pathological developing or surgically induced pathological biomechanics. Kuai et al. analyzed the impact of disc herniation on the kinetics of the spine and lower extremities during everyday activities [44].

Surgical interventions always represent a major intervention in the natural biomechanics of the musculoskeletal system. Thus, several studies on the effects of spinal fusion can be found in the literature [29,71,119]. The resulting kinematic effects of spinal fusion were investigated by Ignasiak et al., who proposed a method for the prediction of a full-body sagittal alignment including reciprocal changes as a reaction to spinal fusion [68].

## 5. Limitations and Challenges

It is in the nature of numerical models that they come with limitations. One of the great challenges is to keep the balance between necessary accuracy and reasonable complexity. This requires not only in-depth knowledge of the object to be modeled but also the corresponding data from experimental studies and the appropriate technical solutions for implementation. During our literature research, we were able to identify several core limitations that could be found in a wide range of MBS models of the spine and the related challenges when it came to addressing these limitations.

### 5.1. Database

Any model can only be as good as its input data. In the context of biomechanical models, this comprises bony geometry, anthropometry, muscle architecture, the mechanical parameters of viscoelastic components and kinematic data. Due to the necessary measurements to determine these parameters, it is currently not possible to build models based on fully consistent datasets. While anthropometric and kinematic data can be determined via noninvasive measures in biomechanics labs, such as marker-based motion capturing, the derivation of bony geometries, muscle architecture, and a detailed distribution of soft tissue usually need medical imaging or are performed in cadaver studies. However, the mechanical properties of viscoelastic components such as ligaments or the IVD can currently only be determined with the help of in vitro studies, which require the isolation of the structure of interest to mount them in respective testing machines. Consequently, these measurements are also usually performed with specimens from cadaver studies and highly dependent on the experimental conditions.

In the past years, more studies including widely individualized models were published [55,59,62]. However, even these models can only offer a limited customization.

In order to obtain consistent data sets for biomechanical models, alternative, noninvasive methods must be developed to determine these parameters in large subject cohorts. Here, the combination of experimental studies, multimodal imaging, and ANNs could be a possible solution to increase the level of model individualization beyond its anthropometric and skeletal characteristics. Thus, the individual mechanical condition of functional components can be evaluated partly on the basis of imaging data. For instance, according to the Pfirrmann scale, a potential degradation of the IVD can be determined via the height and signal intensity from MRI data [120]. Correlating this degradation with the mechanical alteration of IVD [121], this can be used to consider the individual mechanical state of connective tissue, when it is implemented in respective models. Training ANNs with these data will provide large, more diverse datasets for individualized multibody models.

Furthermore, invasive experimental studies on spinal loading for model validation are rare and are not widely feasible due to ethical reasons. Accordingly, even consistently constructed models cannot ultimately be validated against data pertaining to the individual in question. Additionally, the high level of variability in mechanical properties of biological materials as mentioned in Section 3.2, and therefore, the integration of parameters with high standard deviations inevitably leads to models containing inaccuracies. Depending on the complexity of the model, these inaccuracies can accumulate and further blur the generated results. It is necessary to be aware of existing inconsistencies and imprecision when interpreting simulation results in order not to draw incorrect conclusions.

### 5.2. Joint Definition

Intervertebral connections are a complex combination of the IVD, ligaments, facet joints, and articulated capsules. Depending on the applied load, this leads to complicated kinematics in which the instantaneous center of rotation migrates in the course of the motion [122]. However, in the vast majority of spine models, intervertebral joints are simplified to spherical joints allowing three rotational DOFs around a fixed center of rotation. The sensitivity of this assumption has been subject to several in silico studies [23,113,123], indicating that the effect of this assumption on the calculated muscle forces and spinal loading should not be neglected. Detailed modeling requires six degrees of freedom and the consideration of appropriate stabilizing structures, the validity of which depends primarily on the definition of their mechanical parameters (Section 5.1). There are some models to be found in the literature incorporating such detailed representation of intervertebral connection [22], mainly focusing on load sharing in passive structures.

Larger data sets could also help to better understand intervertebral dynamics in order to develop corresponding valid modeling approaches. As already mentioned in Section 5.1, the combination of imaging, machine learning for process automation, and in vitro studies can contribute to progress.

### 5.3. Intra-Abdominal Pressure

The stabilizing influence of intra-abdominal pressure (IAP) on the spine has been widely studied [124,125]. However, only a few MBS models consider its effects [38,60,63,70,77]. In consequence, spinal loads in lifting tasks or the inclination of the upper body are assumed to be overestimated in the MBS modeling of the spine. Arshad et al. observed a decrease of up to 514 N in lumbar compression force and 279 N in global muscle force due to the inclusion of intra-abdominal pressure [38]. These results indicated that it was necessary to consider the effects of IAP to obtain reliable quantitative results on spinal loads.

### 5.4. Muscle Modeling and Muscle Force Estimation

A valid representation of relevant muscles is crucial to gain meaningful findings on the biomechanics of the spine. Most of the models contain a detailed muscle architecture consisting of multiple fascicles spanning between origin and insertion according to the literature. Deploying modeling components, that are usually defined as point-to-point force elements, can lead to nonphysiological lever arms depending on the imposed movement. De Zee’s model used so-called via points to redirect the lines of action of the modeled long muscle fascicles along the rib cage and thus create more realistic lines of action compared to simple straight lines [30]. However, this approach came with an increased computational cost, making it only conditionally suited for a systematic analysis of large participant cohorts.

Another aspect that has to be critically discussed is the applied muscle model. While simple force actuators are considered sufficient for a static investigation, high-dynamic load situations require the consideration of activation and contraction dynamics. This requires an in-depth knowledge of the characteristics of individual muscle morphology such as optimal fiber length, physiological cross-sectional area (PCSA), or pennation angle. Again, the need for subject-specific solutions is evident, as muscle morphology is highly dependent on the individual.

The vast majority of currently published models use a combination of inverse dynamics and static optimization for muscle force calculation. This approach provides a sufficient accuracy in static and quasi-static simulations but is dependent on the defined cost function, constraints, and used algorithm. Most commonly used are criteria for minimum fatigue [126], or the sum of squared muscle strength [127] or activation [34], and the maximum muscle stress is defined as the upper-bound constraint, which is usually set to 100 MPa [32,49,59] to guarantee that equilibrium conditions are met reliably. However, this value does not correspond to a physiological value [49]. Furthermore, SO neglects cocontraction, which incorporates the activation of the antagonist in addition to the agonist stabilizing the respective joint and therefore increasing muscle activation. This is in contradiction to the idea of static optimization, which aims at minimizing the defined cost function (e.g., muscle activation) [128]. In high-dynamic load cases, where the role of cocontraction is more evident, this leads to an underestimation of spinal loading.

One way to address this problem is to use dynamic optimization (DO). In contrast to static optimization, the entire time history of the motion under investigation is taken into account [128]. Integrating the respective criteria in the optimization objective, stabilizing effects such as cocontraction can come into play [25]. However, this method comes with a massive increase of computational cost [129]. Another possibility would be to train models with the help of artificial intelligence. However, such training requires large quantities of data, which is not possible due to the still widely manual and therefore time-consuming process of modeling [128]. Anderson et al. compared both approaches for the simulation of normal gait in 2001, stating that both provided equivalent results for low-dynamic simulations [129]. A similar comparison was made by Morrow et al. for wheelchair propulsion, noticing significant differences in estimated muscle activations [130]. Keeping in mind that wheelchair propulsion comprises higher dynamics than normal gait, these findings indicate that the validity of the chosen approach was largely dependent on the investigated load case.

## 6. Conclusions

Multibody models are a powerful tool to gain insight into the healthy and pathological musculoskeletal system. They can promote a general understanding of the patho-biomechanics of a large set of medical impairments and might even be able to support diagnostics and therapy planning in the future. Although simplifications and assumptions are an integral part of any model, it is essential to look closely at the implications of these assumptions, potential interactions, and possible solutions. Modern technology holds the potential to provide some of these solutions. Thus, artificial intelligence and state-of-the-art medical imaging can provide the necessary extensive data basis to systematically investigate critical parameters to derive appropriate solutions. These technical approaches coupled with a distinct awareness of existing limitations will lead us towards a growing, more profound understanding of musculoskeletal mechanics.

## Figures and Tables

**Table 1 bioengineering-10-00202-t001:** Overview of original models of the musculoskeletal thoracolumbar spine and related modeling methods. Semi-individualized models are those that contain both individualized and generic musculoskeletal components. Joint definitions include potentially assigned constraints.

Reference	Included Segments	Joint Definition	Generic/Indiv.	Passive Force Elements	Muscle Model and Force Estimation	Software	Related Studies
de Zee et al. [30]	Pelvis, sacrum, L1-L5, thorax	3 rot. DOFs (IV)	Generic	-	Act., ID, SO	AnyBody	[33,35,36,39,44,45,47,65]
Christophy et al. [31]	Pelvis, sacrumL1-L5, thorax	3 rot. DOFs (IV)	Generic	-	Hill type	OpenSim	[37,40,41,46,48,49,50,51,52,53]
Bruno et al. [32]	Pelvis, sacrumT1-L5, ribs, sternum, upper limbs, head–neck	3 rot. DOFs (IV)1 rot. DOFs (CV)	Generic	-	Hill type, ID, SO	OpenSim	[38,42,43,54,55,56,57,58]
Ignasiak et al. [33]	Pelvis, sacrum T1-L5, ribs, sternum head–neck	6 rot. DOFs (IV)1 rot. DOFs (CV/CT)3 rot. DOFs (CS I)6 rot. DOFs (CS II-X)	Generic	CS, CT, CV, IV joint (lin.)	Act., ID, FSK [66], SO	AnyBody	[39,67,68]
Lerchl et al. [59]	Pelvis, sacrum, L1-L5, thorax, upper limbs, head–neck	3 rot. DOFs (IV)	Semi-indiv.	Lig. (nonlin.)IVD (nonlin.)	Actuators, ID, SO	Simpack	-
Favier et al. [69]	Lower limbspelvis, sacrum, L1-L5, thorax (3 segments), upper limbs, head–neck	3 rot. DOFs (IV)	Semi-indiv.	Joint (lin.)	Hill type, IK, ID, SO	OpenSim	-
Malakoutian et al. [60]	Pelvis, sacrum, L1-L5, thorax, humeri	6 DOFs (IV)	generic	Joint, IAP	Hill type, FD-assisted SO	AriSynth	[70]
Rupp et al. [61]	Pelvis, sacrum, L1-L5, thorax	6 DOFs (IV)	Generic	Lig. (nonlin.)IVD (nonlin.)	Hill type, FD	In-house	-
Fasser et al. [62]	Pelvis, sacrum, L1-L5, thorax	3 rot. DOFs (IV)	Semi-indiv.	-	Hill type, IK, ID, SO	Matlab	[71]
Bayoglu et al. [72]	Pelvis, sacrum, C1-L5, ribs, sternum, skull (3 segments), shoulder (3 Segments)	3 rot. DOFs (IV)6 DOFs (CS)1 DOF (CV/CT)	Individ.	Joint (lin.)	Act., ID, SO	AnyBody	[73,74,75]
Huynh et al. [63]	Full-body, C1-L5	3 rot. DOFs (IV)	Generic	Lig. (lin.)IVD (lin.), IAP	IK, ID, SO	LifeMOD	[76]
Khurelbaatar et al. [64]	Pelvis, sacrum, C1-L5, ribs, sternum, upper limbs, head	6 DOFs (IV/CS), 3 rot. DOFs (CV)	Semi-indiv. (bones)	Lig. (nonlin.), IVD (nonlin.), CS cartilage (lin.), facet joints	Act., ID, SO	RECURDYN	-
Guo et al. [77]	Pelvis, sacrum, C1-L5, ribs, sternum, upper limbs, head	6 DOFs (IV)	Generic	Lig. (nonlin.), IVD (lin.), facet joints, IAP	Hill type, ALE, FD	OpenSim	-

The definition of the abbreviations can be found at the end of this article.

**Table 2 bioengineering-10-00202-t002:** Overview of representative studies using available original models to address methodological or biomechanical research questions.

Study	Focus	Modifications	Original Model
Actis et al. [48]	Methodological Validation for flexion, extension, lateral bending, axial rotation for participants with and without transtibial amputation	model extension by lower body [110], muscle strength [32], and body mass distribution [86] inclusion of experimental protocol for EMG and kinematic data collection	[31]
Arshad et al. [38]	Biomechanical Influence of spinal rhythm and IAP on lumbar loads during trunk inclination	Adapted spinal rhythm, inclusion of ligaments, IVD, and IAP	[30]
Arx et al. [83]	Biomechanical Lumbar loading during different lifting styles	Integration of measured kinematic data	[32]
Banks et al. [58]	Biomechanical Comparison of static and dynamic vertebral loading during lifting patient-specific models in an older study population	CT-based individualization and integration of patient-specific kinematic data	[32]
Bassani et al. [45]	Methodological Model validation for various loading tasks via spinopelvic rhythm and IDP according to [4]	Integration of kinematic data	[30]
Bassani et al. [47]	Biomechanical Effect of spinopelvic sagittal alignment on lumbar loads	Variation of spinal alignment based on four parameters	[30].
Bayoglu et al. [75]	Methodological Sensitivity of muscle and IV disc force computations to variations in muscle attachment sites	Variation of the location of muscle insertion	[72]
Raabe et al. [40]	Biomechanical Jogging biomechanics	Combination with full-body model by [111]	[31]
Beaucage-Gauvreau et al. [49,50,51]	Biomechanical Effects of lifting techniques on lumbar loads	Adjust all spinal joints with 3 DOFs and inclusion of kinematic data from motion capturing during lifting	[31,40]
Burkhart et al. [54]	Methodological Reliability of optoelectronic motion capturing for subject-specific spine model generation	Combination with model of lower limbs [110]	[32]
Malakoutian et al. [70]	Methodological Effect of muscle parameters on spinal loading	Variation of biomechanical parameters of paraspinal muscles	[60]
Senteler et al. [41]	Methodological Joint reaction forces for flexion and lifting	Combination with models of upper limbs and neck, IV joints set to 6 DOFs, added passive lin. joint stiffness	[31]
Meng et al. [37]	Methodological Force-motion coupling in 6-DOF joint	6 DOFs (IV), added 6-DOF stiffness	[31]
Molinaro et al. [52]	Biomechanical Effects of throwing technique solid waste collection occupation on lumbar loads	Incorporation of collected kinematics and EMG data, EMG-assisted muscle force estimation and SO	[49]
Schmid et al. [56]	Methodological Validation of a thoracolumbar model for children and adolescents	Combination with model of the lower limbs [112], scaling to anthropometry of children and adolescents	[32]
Schmid et al. [57]	Methodological Feasibility of a skin-marker based method for spinal alignment modeling	Reduction of muscle architecture, implementation of skin-marker derived alignment	[56]
Wang et al. [84]	Methodological Implementation of a physiological FSU	Adaption of IV joints to represent passive properties of a physiological FSU	[32]
Overbergh et al. [55]	Methodological Workflow for generation of an image-based (CT), subject-specific thoracolumbar model of spinal deformity	Addition of kinematic coupling constraints, personalization of bone geometries, alignment, IV joint definitions and kinematics	[32]
Han et al. [36]	Methodological Effect of centers of rotation on spinal loads and muscle forces in total disc replacement of lumbar spine	Ligaments and facet joints added, altering location of CoR	[30]
Zhu et al. [46]	Biomechanical Effects of lifting techniques on lumbar loads	Combining with models of upper and lower limbs, 6-DOF IV joint, integration of a customized marker set	[31]
Kuai et al. [44]	Biomechanical Influence of disc herniation on kinematics of the spine and lower limbs	Integration of kinematic data from patients with lumbar disc herniation	[30]
Senteler et al. [113]	Methodological Sensitivity of intervertebral joint forces to CoR location	Altering location of CoR	[41]

## Data Availability

Not applicable.

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
