# Peer review of "Multibody Models of the Thoracolumbar Spine: A Review on Applications, Limitations, and Challenges"

_bioengineering, 2023, doi:10.3390/bioengineering10020202_

Round 1

Reviewer 1 Report

The paper performs an extensive review of the literature regarding biomechanics of the thoracolumbar spine that uses multibody models, which are understood as mechanisms composed by rigid bodies connected by kinematic joints and passive/active force elements. Aspects as kinematics, scaling and muscle force estimation are considered (among others) providing fairly large and detailed lists of previous works, clearly organized.  The paper is very well written and I think that it contains very useful information for the biomechanics community.

Comments:

- - I suggest to add the word "kinematics" to the title of section 3.1: "General model setup and kinematics"

- Several aspects of the existing models have been addressed (passive (visco)elastic components, scaling and individualization, etc), but I miss some comments regarding specific aspects of the numerical solution of these systems. These aspects may considerably affect the computational cost, accuracy and stability of the solution.  For instance:

-- Type of coordinates (absolute/relative; minimum/redundant, etc.)

-- Modelling of constraints.  Constraints appear in the definition of joints (articulations) or may be necessary due to the redundancy of the selected type of coordinates (e.g, if the coordinates are cartesian coordinates of selected points in the bodies).

-- In case of forward dynamics, time integration (type, order, constant/adaptive time step-size)

- In the last decades, the concept of "multibody system" has been generalized a little bit, comprising both rigid and deformable bodies connected by kinematical joints. The deformable bodies could be continuuum-3D or special structural elements, such as beams and shells.  It would be interesting to say something about the existence (or absence) of this type of models.

- The identification and definition of realistic boundary conditions is often a very challenging issue in the numerical solution of the dynamics of multibody systems. It would be interesting to say something about this issue.

As a minor formal issue, the list of abbreviations in page 12 is very useful; nevertheless, it would improve readability to have each abbreviation explained at the first appearance in the main text. For instance, the meaning of FEM is clear from line 22 of page 1, but SO is first used in the text without explanation in line 400 in page 11.

Author Response

Revision
Dear Reviewers,

Thank you for giving us the opportunity to submit a revised draft of the manuscript entitled “Multibody Models of the Thoracolumbar Spine: A Review on Applications, Limitations and Challenges” for publication in the special issue “Biomechanics-Based Motion Analysis” in Bioengineering. We appreciate the time and effort invested by you and the Reviewers to provide valuable feedback on our manuscript and are grateful for your insightful comments. All of the comments provided enabled us to improve the quality and the content of the manuscript. We have carefully considered all of the suggestions made by the Reviewers and addressed and incorporated each one of them. All changes are highlighted within the manuscript, while our point-by-point responses to the Reviewers’ comments and concerns are provided below.

We hope that the revised manuscript and our accompanying responses are sufficient to make our manuscript suitable for publication in Bioengineering.

Yours sincerely,

Tanja Lerchl & Jan S. Kirschke

Reviewer 1:

R1/1- I suggest to add the word "kinematics" to the title of section 3.1: "General model setup and kinematics"
Response: Thank you for this comment. We added “kinematics” to the title and extended the respective paragraph in the section.

R1/2 - Several aspects of the existing models have been addressed (passive (visco)elastic components, scaling and individualization, etc), but I miss some comments regarding specific aspects of the numerical solution of these systems. These aspects may considerably affect the computational cost, accuracy and stability of the solution. For instance:

a) Type of coordinates (absolute/relative; minimum/redundant, etc.)
Response: Thank you for this important remark. This is undoubtedly important considering the simulation-specific aspects mentioned. In our own models, we use relative minimum coordinates to define joints and kinematics. The authors of the reviewed publications don’t address this issue explicitly. However, it is common in biomechanical analyses to define model kinematics and setup using relative minimum coordinates. An exception here are inverse kinematic approaches, in which the kinematics of the end link of the kinematic chain are usually given in absolute terms. We added this aspect to the paragraph on joint kinematics.

(b) Modelling of constraints. Constraints appear in the definition of joints (articulations) or may be necessary due to the redundancy of the selected type of coordinates (e.g, if the coordinates are cartesian coordinates of selected points in the bodies).
Response: Thank you for this comment. The reviewed article mentioned constraints primarily in the context DOF definition in the intersegmental joints. We added this information in the respective paragraph.

(c) In case of forward dynamics, time integration (type, order, constant/adaptive time step size)
Response: Thank you for this remark. We agree, that this would be a valuable information, especially for dynamic load cases.. However, in the reviewed literature, only few forward dynamic models could be found and unfortunately, no detailed information on time integration specifications was given.

R1/3 - In the last decades, the concept of "multibody system" has been generalized a little bit, comprising both rigid and deformable bodies connected by kinematical joints. The deformable bodies could be continuuum-3D or special structural elements, such as beams and shells. It would be interesting to say something about the existence (or absence) of this type of models.
Response: Thank you for this important comment. We agree, that the existence of these models represents an interesting development in musculoskeletal modeling and added a respective paragraph in the introduction. However, most models for clinical or biomechnical analyses, use classic rigid body models, which is why we focussed on these models for the scope of this review.

R1/4 - The identification and definition of realistic boundary conditions is often a very challenging issue in the numerical solution of the dynamics of multibody systems. It would be interesting to say something about this issue.
Response: Thank you for this remark. Regarding kinematic boundary conditions, we added respective information in Section 3.1. Regarding boundary constraints due to biological constrains, such as the intraabdominal pressure, please refer to the respective section 5.3 on page 11. In case I misunderstood your remark, I’d be happy to address it again with additional information.

R1/5 - As a minor formal issue, the list of abbreviations in page 12 is very useful; nevertheless, it would improve readability to have each abbreviation explained at the first appearance in the main text. For instance, the meaning of FEM is clear from line 22 of page 1, but SO is first used in the text without explanation in line 400 in page 11.
Response: Thank you for this comment. We tried to explain all abbreviations at first appearance in the text, but obviously forgot a few. We added the missing abbreviations.

Reviewer 2 Report

A literature review is presented summarizing the current knowledge about multibody models of the thoracolumbar spine. Overall, this review is nicely written and presented in a clear and concise manner, while containing the pertinent literature. There are, however, some missing aspects which should be added and some issues which have to be clarified prior to publication of this manuscript.

Introduction

Lines 20-21: What about experimental (in vitro) models?

Methods

Lines 45-46: Did the systematic literature search follow the PRISMA guidelines? If not, how was reproducibility of the literature search ensured?

Line 47: Are there not brackets missing before the keyword ‘multi’ and after the keyword ‘body’, thus leading to the search term ‘spine model multi musculoskeletal’ instead of ‘spine model musculoskeletal’?

Line 48: Why is there such a large discrepancy between the number of search results between PubMed and Scopus? One would expect numbers on approximately the same scale. Could this be related to the search term? (How) were duplicates removed?

Line 51: Why were solely exclusion criteria defined and no inclusion criteria? Could this approach have caused a limited search output, e.g. for studies which have used multiple models?

Line 54: What does ‘models without model incorporation’ exactly mean? Can the authors give an example?

Multibody modeling of the Healthy Spine

Line 65: ‘reviled’ -> ‘revealed’

Lines 71-73: What does ‘semi-individual’ exactly mean in Table 1? Were there generic elements used? This should be further explained, e.g. in the Table caption.

Lines 158-159: There are usually two different strategies of implementing material parameters into numerical models, which should be further outlined and discussed here: (1) Direct implementation of material parameters from literature (e.g. Pintar et al. 1992, J Biomech 25, 1351–1356; Myklebust et al. 1988, Spine 13, 526-531; etc.) or (2) indirect implementation of material properties by reproducing experimental testing, optimally by reproducing stepwise resection of structural elements (e.g. Heuer et al. 2007, J Biomech 40, 271–280 for lumbar spine; Wilke et al. 2020, Eur Spine J 29, 179-185 for thoracic spine; etc.).

Limitations and Challenges

Lines 332-350: Another limitation of the validation of numerical models using experimental data is the problem of large standard deviations of single experimental parameters, which might be accumulated with increased number of parameters, i.e. increasing complexity of the numerical model.

Lines 340-345: Is it really plausible that biomechanical data can be derived solely from imaging without performing experimental testing as it is implied in this passage? Such a statement should be made with caution without having a proof of concept. At least, such artificial neural networks based on imaging have to be trained by experimental testing in order to provide realistic results.

Line 390: ‘pinnation’ -> ‘pennation’

Conclusion

Lines 421-422: ‘Multibody models (…) can support diagnostics and treatment planning …’ This idea could become true in the future but is currently not feasible, which should be stated here.

Author Response

Revision

Dear Reviewers,

Thank you for giving us the opportunity to submit a revised draft of the manuscript entitled “Multibody Models of the Thoracolumbar Spine: A Review on Applications, Limitations and Challenges” for publication in the special issue “Biomechanics-Based Motion Analysis” in Bioengineering. We appreciate the time and effort invested by you and the Reviewers to provide valuable feedback on our manuscript and are grateful for your insightful comments. All of the comments provided enabled us to improve the quality and the content of the manuscript. We have carefully considered all of the suggestions made by the Reviewers and addressed and incorporated each one of them. All changes are highlighted within the manuscript, while our point-by-point responses to the Reviewers’ comments and concerns are provided below.

We hope that the revised manuscript and our accompanying responses are sufficient to make our manuscript suitable for publication in Bioengineering.

Yours sincerely,

Tanja Lerchl & Jan S. Kirschke

Reviewer 2:

Introduction

R2/1- Lines 20-21: What about experimental (in vitro) models?
Response: Thank you for this comment. We added a statement on in vitro studies to the introduction.

Methods

R2/2- Lines 45-46: Did the systematic literature search follow the PRISMA guidelines? If not, how was reproducibility of the literature search ensured?
Response: Thank you for this remark. We have followed the PRISMA guidelines for data collection to the extent, that the generated results are reproducible.

R2/3- Line 47: Are there not brackets missing before the keyword ‘multi’ and after the keyword ‘body’, thus leading to the search term ‘spine model multi musculoskeletal’ instead of ‘spine model musculoskeletal’?
Response: Thank you for this very important comment. We agree, that we made a mistake here. We repeated the search considering your comment. While the change in search terms didn’t affect the results on PubMed, the results on Scopus increased to 2032. Excluding unsuitable subject areas left 1302 documents. Relating to your following comment, this represents a resulting number in the same scale as the results on PubMed. We removed
duplicates by detecting identical PubMed IDs and titles (698), and subsequently, filtered the resulting documents according to our inclusion and exclusion criteria. Only few new publications remained. Since these publications did not include relevant new aspects, we only cited them in pertinent passages. We changed the paragraph on methods accordingly. Please note, that the change tracking didn’t work for the list of inclusion criteria.

R2/4- Line 48: Why is there such a large discrepancy between the number of search results between PubMed and Scopus? One would expect numbers on approximately the same scale. Could this be related to the search term? (How) were duplicates removed?
Response: Thank you for this remark. Please refer to our previous response on that matter.

R2/5- Line 51: Why were solely exclusion criteria defined and no inclusion criteria? Could this approach have caused a limited search output, e.g. for studies which have used multiple models?
Response: Thank you for this comment. We considered studies with multiple models during our initial search. However, we agree, that the respective paragraph might be confusing. We extended it by our inclusion criteria.

R2/6- Line 54: What does ‘models without model incorporation’ exactly mean? Can the authors give an example?
Response: Thank you for this remark. This is a typo. It’s supposed to say “models without muscle incorporation”. We corrected the sentence. Multibody Modeling of the Healthy Spine

R2/7- Line 65: ‘reviled’ -> ‘revealed’
Response: Thank you for this remark. We corrected the typo.

R2/8- Lines 71-73: What does ‘semi-individual’ exactly mean in Table 1? Were there generic elements used? This should be further explained, e.g. in the Table caption.
Response: Thank you for this comment. We added the explanation to the caption.

R2/9- Lines 158-159: There are usually two different strategies of implementing material parameters into numerical models, which should be further outlined and discussed here: (1) Direct implementation of material parameters from literature (e.g. Pintar et al. 1992, J Biomech 25, 1351–1356; Myklebust et al. 1988, Spine 13, 526-531; etc.) or (2) indirect implementation of material properties by reproducing experimental testing, optimally by reproducing stepwise resection of structural elements (e.g. Heuer et al. 2007, J Biomech 40, 271–280 for lumbar spine; Wilke et al. 2020, Eur Spine J 29, 179-185 for thoracic spine; etc.).
Response: Thank you for this remark. We extended the paragraph by the respective information.

Limitations and Challenges

R2/10- Lines 332-350: Another limitation of the validation of numerical models using experimental data is the problem of large standard deviations of single experimental parameters, which might be accumulated with increased number of parameters, i.e. increasing complexity of the numerical model.

Response: Thank you for this important comment. This in undoubtedly an important issue in biomechanical modeling. We included a paragraph on that topic.

R2/11- Lines 340-345: Is it really plausible that biomechanical data can be derived solely from imaging without performing experimental testing as it is implied in this passage? Such a statement should be made with caution without having a proof of concept. At least, such artificial neural networks based on imaging have to be trained by experimental testing in order to provide realistic results.

Response: Thank you for this important remark. We agree, that ANNs need to be trained by data from experimental studies in order to generate reliable results. The paragraph in question is intended to point out the potential of ANNs in combination with multimodal imaging and findings from experimental data to provide large, more diverse datasets for individualized multibody models not only for body segmentation, but also regarding mechanical properties of functional components. For instance, ANNs can be trained to detect IVD degradation based on parameters derived from imaging data (e.g. disc height or signal intensity, grading according to Pfirrmann), which further can be correlated with changes in mechanical properties. However, this is not stated clearly. We extended the paragraph and added respective literature.

R2/12- Line 390: ‘pinnation’ -> ‘pennation’
Response: Thank you for this remark. We corrected the typo.

Conclusion

R2/13- Lines 421-422: ‘Multibody models (…) can support diagnostics and treatment planning …’ This idea could become true in the future but is currently not feasible, which should be stated here.
Response: Thank you for this remark. We agree, that our formulation did not make clear, that this statement refers to the potential of biomechanical modelling, rather that its current feasible use. We reformulated this sentence to emphasize this aspect.

Round 2

Reviewer 2 Report

Thank you for this excellent literature review. For future systematic reviews, I would recommend using and stating PRISMA guidelines.